# Influence of the Thermal State of Vehicle Combustion Engines on the Results of the National Inventory of Pollutant Emissions

**Katarzyna Bebkiewicz** [1], **Zdzisław Chłopek** [2], **Hubert Sar** [2,*], **Krystian Szczepański** [3] and **Magdalena Zimakowska-Laskowska** [1]

[1] National Centre for Emissions Management (KOBiZE), Institute of Environmental Protection-National Research Institute, 132/134 Chmielna Str., 00-805 Warsaw, Poland; katarzyna.bebkiewicz@kobize.pl (K.B.); magdalena.zimakowska-laskowska@kobize.pl (M.Z.-L.)

[2] Institute of Vehicles and Construction Machinery Engineering, Warsaw University of Technology, 84 Narbutta Str., 02-524 Warsaw, Poland; zdzislaw.chlopek@pw.edu.pl

[3] Institute of Environmental Protection-National Research Institute, 5/11D Krucza Str., 00-548 Warsaw, Poland; krystian.szczepanski@ios.edu.pl

\* Correspondence: hubert.sar@pw.edu.pl; Tel.: +48-22-234-8545

**Abstract:** The article presents the results of studies on the influence of the thermal state of vehicle combustion engines on pollutant emissions. This influence was analyzed based on data from Poland's inventory of pollutant emissions for the years 1990–2017. The results show that during engine warm-up, carbon monoxide emission constitutes the largest share (up to 50%) in the national annual total emission. Volatile organic compounds are next in the ranking, whereas the share of nitrogen oxides is the lowest (less than 5%). Under the model traffic conditions, close to those in Poland's cities in winter, simulation tests regarding additional pollutant emissions from passenger cars during engine warm-up were also carried out. As a result of the cold-start emissive behavior of internal combustion engines, emissions of carbon monoxide and volatile organic compounds showed a considerably greater impact on national pollutant emission, as compared to carbon dioxide, nitrogen oxides and particulate matter. This is particularly evident for the results of the inventory of pollutant emissions from road transport.

**Keywords:** pollution; inventory of pollutant emissions; methodology of pollutant research; traffic congestion; exhaust fumes

## 1. Introduction

Technological progress has allowed the introduction of innovative solutions in the construction of internal combustion engines and large reductions in exhaust gas emissions. The advancements are acknowledged in evolving law regulations on the protection of the environment against negative impacts of automotive internal combustion engines, also in terms of the elaboration of acceptable limits for criterion values regarding pollutant emissions from the specific distance emission (g/km) for passenger cars, light trucks, motorcycles and mopeds [1], as well as the specific brake emission (g/(kW·h)) from trucks and buses [2]. Accordingly, the inventory of pollutant emissions from road transport has evidently shown progress in emission reductions [3–8], regardless of further significant intensification of road transport (the dynamic growth in the number of means of transport and their annual mileage) [6–8]. Notwithstanding considerable progress made, further solutions to reduce vehicle pollutant emissions are being sought after. One of them is undertaking an effort to reduce pollutant emissions from fuel combustion during engine warm-up prior to reaching a stabilized thermal state.

It is well known that fuel consumption and pollutant emissions increase during the engine warm-up process due to the following factors [9–11]:

- low efficiency of the catalytic converters due to too low temperatures of engine systems,
- less efficient combustion process under low engine temperatures,
- supplying cylinders with a richer mixture in the case of warmed-up engines.

There is a comprehensive literature available on the subject of pollutant emissions from internal combustion engines during warm-up, e.g., [1,9–22].

Numerous published results refer to empirical studies on pollutant emissions from internal combustion engines at different thermal states. In [14], the characteristics of particulate matter emitted from a two-cylinder diesel engine were examined. It was observed that particulate matter emission was highly sensitive to the thermal status of the engine tested. In [19], the effect of the start-up temperature on the pollutant emissions was assessed from a compression-ignition engine, under start temperatures: $-7\,°C$ and $20\,°C$, with the use of the type-approval test NEDC (New European Driving Cycle) and WLTC (Worldwide Harmonized Light Vehicles Test Cycle). The results obtained confirmed an explicit increase in the emissions of carbon monoxide—CO, hydrocarbons—HC and nitrogen oxides—NOx, as well as in fuel consumption. In the study [20], pollutant emissions from a compression-ignition engine were tested with the use of the NEDC and Federal Test Procedure (FTP-75). The results obtained were similar to those in [19]. In addition, a significant increase in particulate matter emission was also seen during engine cold-start. The study [23] focused on particulate matter with the diameter below 100 nm (nanoparticles). The results obtained showed a clear increasing tendency in solid particle size under cold-start conditions, both for compression-ignition and spark-ignition engines with direct fuel injection. In [21], the effect of increasing the temperature of the air filling the compression-ignition engine cylinders on pollutant emissions under cold-start conditions was examined. The proposed method was highly effective—the significant reduction in pollutant emissions was found under the cold-start condition. In [11], the emission of pollutants from spark-ignition engines (both multipoint- and direct-injection) was examined under temperatures from $-7\,°C$ to $30\,°C$. A higher sensitivity of pollutant emissions to ambient temperature was found for multipoint-injection engines. In [22], pollutant emissions were examined during the cold-start of a spark-ignition engine used in small heat and power generating plants. In [18], the parameters of the combustion process were tested with the use of the single-cylinder diesel engine with an optical combustion chamber. The test results confirmed the impact of a low start-up temperature on the parameters of the combustion process.

Several studies previously focused on the simulation tests of pollutant emissions from engines under cold-start conditions [1,9,10] and the inventory of pollutant emissions [1,5]. The study [10] presents an analysis of the results of the simulation tests on pollutant emissions from spark-ignition engines during cold-start, depending on: ambient temperature, the time of vehicle standstill before start-up, the length of distance covered by the vehicle after start-up and the vehicle motion model. The study [9], carried out with the use of INFRAS AG software [17] (also used in the study [10]), presents the results of the simulation tests on the emission of organic compounds from automotive combustion engines at the stage of engine warm-up after the start-up. The study [1] presents the results of simulation tests on pollutant emissions at the cold-start of spark-ignition engines for the structure of passenger cars and light trucks in Australia. This study was carried out with the use of COPERT software [24]. The studies [1,5] present the results on the inventory of pollutant emissions associated with the cold-start of spark-ignition engines. The study [5] presents the results of greenhouse gas (GHG) emissions. The study [1] examined the emission of carbon monoxide, organic compounds and nitrogen oxides, as well as fuel consumption during engine warm-up. COPERT software was used for the inventory of pollutant emissions.

Several studies previously addressed the methods for reducing pollutant emissions and fuel consumption during engine cold-start. In [12], test results on a compression-ignition engine with the platinum-coated glow plug were presented. Measurable effects

were achieved in reducing the emissions of carbon monoxide, hydrocarbons and particulate matter, as well as fuel consumption.

The large interest in pollutant emissions and fuel consumption under cold-start conditions for internal combustion engines proves that this issue has gained great interest in the area of construction and the operation of internal combustion engines.

In general, total fuel consumption intensity and pollutant emission intensity can be represented by the D set consisting of two elements: the total fuel consumption intensity and the total pollutant emission intensity:

$$\boldsymbol{D} = [G_F, \boldsymbol{E}] \tag{1}$$

where:

$G_F$—total fuel consumption intensity,
$\boldsymbol{E}$—total pollutant emission intensity—a set of which elements are the emission intensity of an individual pollutant.

Additional fuel consumption and pollutant emissions are modeled, consistent with [10,13,17]:

- ambient temperature—$t_a$,
- rest time before starting the engine—$\tau$,
- distance traveled by the vehicle after start-up—$l$,
- vehicle motion conditions after engine start-up, as described by the model of vehicle movement—$v(t)$.

Additional factors of fuel consumption intensity and pollutant emission intensity are presented as a set:

$$\boldsymbol{D_w} = [G_{Fw}, \boldsymbol{E_w}] \tag{2}$$

where:

$G_{Fw}$—additional fuel consumption intensity during warm-up of the internal combustion engine,
$\boldsymbol{E_w}$—additional pollutant emission intensity during warm-up of the internal combustion engine.

Additionally, fuel consumption intensity and pollutant emission intensity are modeled as an operator [10,13]:

$$\boldsymbol{D_w} = \boldsymbol{B}[t_a, \tau, l, v(t)] \tag{3}$$

There are available models that enable the estimation of additional fuel consumption and pollutant emissions, among others: INFRAS AG [17] and COPERT [24] software.

The emission of pollutants is most sensitive during engine warm-up, especially in the case of spark-ignition engines and—to a much lesser extent—in compression-ignition engines. For this reason, and due to the nature of the work of heavy vehicles, additional increases in fuel consumption and pollutant emission were determined with the use of INFRAS AG and COPERT 5 only for passenger cars and light trucks, equipped with either spark-ignition or compression-ignition engines [24].

## 2. Methodology

At first, the analyses were performed on the pollutant emissions from road transport in Poland for the years 1990–2017, based on the inventory prepared by the National Centre for Emissions Management (KOBiZE) [7,8], in reference to the 2006 IPCC Guidelines for National Greenhouse Gas Inventories [25] and the EEA/EMEP Emission Inventory Guidebook 2019 [15]. COPERT 5 software is currently used to determine annual pollutant emissions from road transport at a national level.

The KOBiZE center, which represents Poland, is one of the world's leaders in the field of inventories of pollutants. This is confirmed by the fact that Poland was awarded the TFEIP (Task Force on Emission Inventories and Projections) under the LRTAP (Convention on Long-Range Transboundary Air Pollution) meeting in 2021 for the best inventory in the "Transparency" category. This fact confirms that the results of the inventory studies

of pollutant emissions from road transport in Poland are representative of economically developed countries.

In the COPERT 5 software, data characterizing the traffic conditions of vehicles in cumulative vehicle categories are entered for:

- passenger cars,
- light commercial vehicles,
- heavy duty trucks,
- urban buses,
- coaches,
- the L-category: motorcycles, mopeds, quads, ATVs (all-terrain vehicles) and micro-cars.

The division of vehicles into categories according to the COPERT 5 software is also presented in Supplementary Materials.

In the model of pollutant emission from road vehicles, it is assumed that the pollutant emission intensity related to the use of road vehicles with internal combustion engines is the sum of the pollutant emission intensity for the states:

- internal combustion engine heated to a stabilized temperature,
- combustion engine heating up,
- evaporation of fuel from the car's fuel system.

Another assumption concerns traffic conditions. It is assumed that the emission of pollutants is the sum of pollutant emissions under model traffic conditions:

- in cities,
- outside cities,
- on highways and expressways.

In the inventory of pollutant emissions, data on the intensity of vehicle use are introduced in elementary categories due to:

- purpose of road vehicles,
- contractual size of road vehicles and their propulsion engines,
- properties of road vehicles and their propulsion engines due to, inter alia, the emission of pollutants,
- fuel for internal combustion engines applied in road vehicles,
- technical level of road vehicles and their propulsion engines.

These data are the number and annual mileage.

According to the EMEP/EEA 2019 guidelines, on which COPERT is based, emissions related to engine heating are determined for passenger cars and light trucks.

The parameters that are entered for elementary vehicle categories are:

- average speed in model traffic conditions,
- share of the distance traveled by the vehicle in model traffic conditions in the total length of the road.

For the purposes of modeling pollutant emissions related to engine heating, the following data shall be taken:

- distance traveled by the vehicle while the engine is warming up,
- extreme average temperatures during the month.

To determine the data, official information comes from, inter alia, the Central Statistical Office in Poland and the Central Register of Vehicles and Drivers. In addition, reports of scientific institutes in Poland are used, including the Motor Transport Institute, Road and Bridge Research Institute, Automotive Industry Institute, BOSMAL Automotive Research and Development Institute.

A detailed set of data adopted each year is included in official reports [7,8].

According to the knowledge of the authors of Polish pollutant emission inventory reports, participating in meetings of the authors of reports from individual Member States, the data adopted in Poland regarding the nature of vehicle traffic are similar to the data adopted in other countries. Of course, the difference occurs in the case of extreme values

of the average ambient temperature per month—these values in Poland come yearly from the official information from the Central Statistical Office. These temperatures in Poland are close to the values adopted in Germany, the Czech Republic, Slovakia, Austria and Switzerland.

Due to the method of collecting data for the inventory of pollutant emissions in Poland, it can be assumed that the obtained results are representative for other European economically developed countries.

In our analyses the following are substances that cause the greenhouse effect intensification and can be hazardous to living organisms:

- carbon monoxide—CO,
- non-methane volatile organic compounds—NMVOC,
- nitrogen oxides NOx: nitrogen dioxide—$NO_2$,
- particulate matter—PM,
- carbon dioxide—$CO_2$.

The analyses for the years 1990–2017 regarded:

- national annual emission of pollutants (total),
- annual emission of pollutants during internal combustion engine warm-up,
- annual emission of pollutants from engine heated to a stable temperature,
- shares of pollutant emissions during cold-start in total annual pollutant emission.

The second step of the study comprised the simulation tests, conducted with the use of INFRAS AG software.

In this software, the additional emission of pollutants related to the start-up of hot engines is taken into account only for passenger cars and light trucks, because heavy vehicles are predominantly equipped with compression-ignition engines, for which the emission of pollutants due to the start-up of a hot engine is much smaller than in the case of spark-ignition engines. Besides, for heavy vehicles, the nature of their use is completely different—long-term use of these vehicles after starting an unheated engine is typical. Due to the low share in the total emissions of motorcycles and mopeds, also in this case cold starts are not taken into account.

Simulation tests were performed for passenger cars with spark-ignition engines (otherwise compression-ignition engines) under the following model conditions:

- ambient temperature: 0 °C,
- the distance traveled by the vehicle after engine start-up: (3 ÷ 4) km,
- parking time before starting the engine: more than 12 h,
- vehicle movement conditions after engine start-up: described by the urban traffic model for an average speed of about 37 km/h.

The research was carried out for the environmental conditions (offered in the software) corresponding to the conditions in Germany, and therefore representative also for Poland.

The choice of these conditions is substantively justified. An ambient temperature of 0 °C was selected, as the relatively low temperature has a decisive influence on the additional emission of pollutants. At the same time, in Poland, the ambient temperature lower than 0 °C is rare. The average speed was assumed to be equal to the average value for passenger cars in cities, adopted in the reported pollutant emission inventory in the COPERT software. Such results of the average speed in cities result from research of motor vehicle traffic in Poland, conducted, among others, by the police. The idle time before starting a cold engine corresponds to the unfavorable situation of cooling the vehicle during the night standstill. The length of the road traveled by the vehicle after starting corresponds to the average road in the city, during which the engine warms up.

Only passenger cars were used in the research for two reasons. First, the emissions of pollutants from passenger car engines are more sensitive to the thermal state of the engine than in light passenger car engines, because the passenger car category has a much greater share of spark-ignition engines than the light trucks category, and the emission of pollutants is more sensitive to the thermal state of the engine for spark-ignition engines

than for compression-ignition engines. Secondly, also including light truck engines would significantly increase the volume of the article.

The following were determined using the INFRAS AG software for the adopted model traffic conditions:

- additional emission of pollutants during engine heating for passenger cars of Euro 0–Euro 6 categories—a software procedure in accordance with Formula (3),
- specific distance emissions of pollutants from heated engines for passenger cars of Euro 0–Euro 6 categories,
- total emission of pollutants from engines for passenger cars of Euro 0–Euro 6 categories in accordance with the Formula (4),
- share of additional pollutant emissions in the time of engine heating in the total pollutant emission.

The specific distance emissions for heated engines in the INFRAS AG software are determined as a function of the average speed.

Therefore, the total pollutant emissions are the sum of:

$$\mathbf{m_T} = \mathbf{m_c} + \mathbf{m_h} = \mathbf{m_c} + \mathbf{b_h} \tag{4}$$

*(The specific distance emission of pollutants is a derivative of pollutant emissions in relation to the distance traveled by the vehicle.)*

where:

$\mathbf{m_T}$—emission of pollutants from engines in model car traffic conditions,
$\mathbf{m_c}$—additional emission of pollutants during engine heating in model car traffic conditions,
$\mathbf{m_h}$—emission of pollutants from engines heated in model conditions of car traffic,
$\mathbf{b_h}$—specific distance emissions of pollutants from engines heated in model car traffic conditions—a set of which elements are specific distance emissions of individual pollutants.

## 3. Results

Detailed information on the evolution of the structure of vehicles used in Poland in the period under consideration, i.e., in the years 1990–2017, and the nature of their operation can be found in the official reports of KOBiZE. Selected elements of this information are also found in publications, e.g., in [5,26,27].

The results of the research using the results of the inventory of pollutant emissions from road transport in Poland in the years 1990–2017 with the use of COPERT 5 software are shown in Figures 1–6.

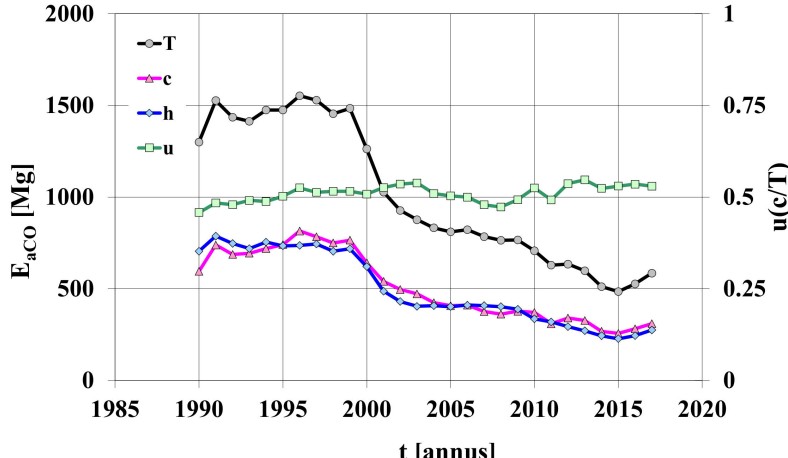

**Figure 1.** Annual carbon monoxide emission—$E_{aCO}$: national (total)—T, from car engines at cold-start—c, from car engines at warm-up—h and the share of additional emissions during engine heating in total emissions—(c/T).

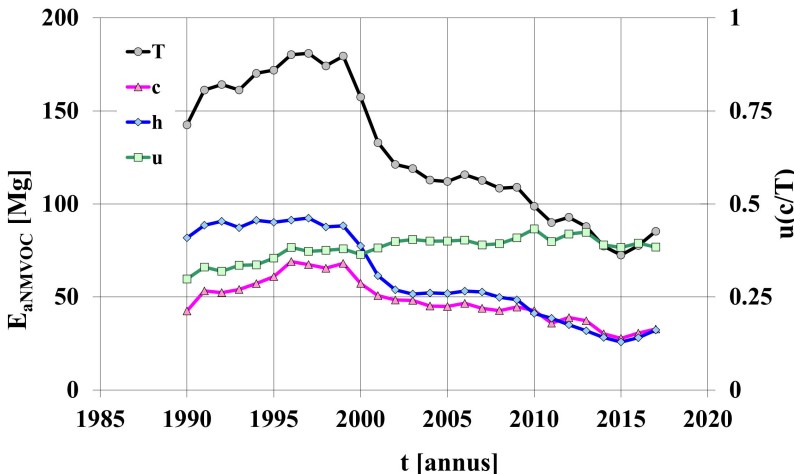

**Figure 2.** Annual emission of non-methane volatile organic compounds: $E_{aNMVOC}$: national (total)—T, from car engines at cold-start—c, from car engines at warm-up—h and the share of additional emissions during engine heating in total emissions—u(c/T).

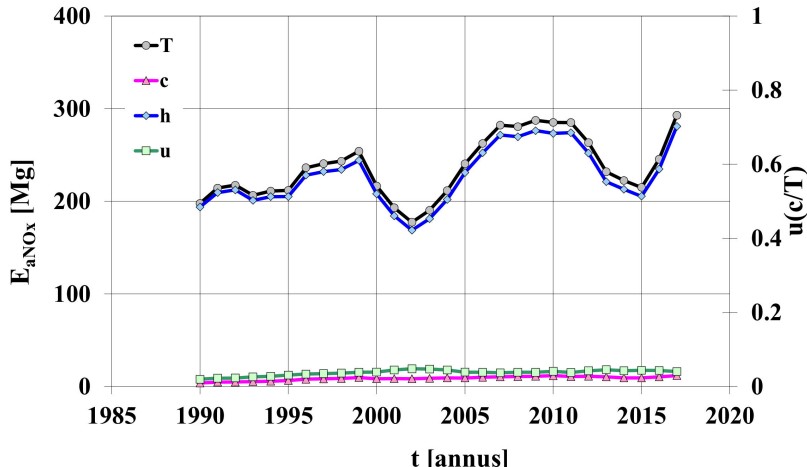

**Figure 3.** Annual emission of nitrogen oxides—$E_{aNOx}$: national (total)—T, from car engines at cold-start—c, from car engines at warm-up—h and the share of additional emissions during engine heating in total emissions—u(c/T).

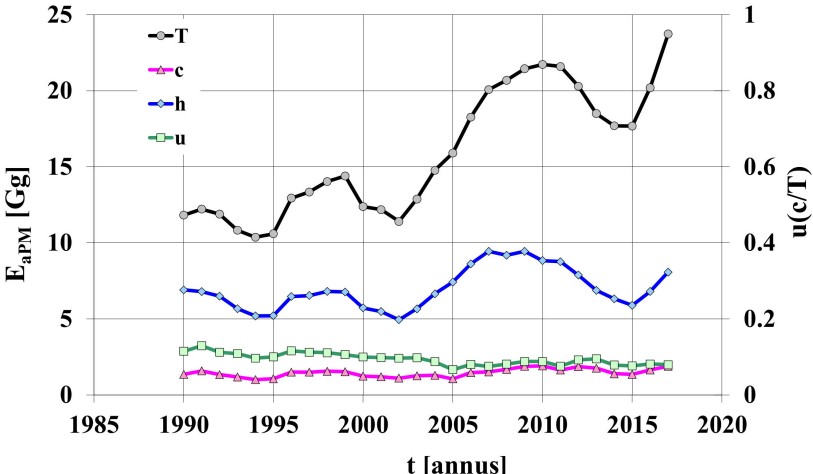

**Figure 4.** Annual emission of particulate matter—$E_{aPM}$: national (total)—T, from car engines at cold-start—c, from car engines at warm-up—h and the share of additional emissions in the time of engine heating in total emissions—u(c/T).

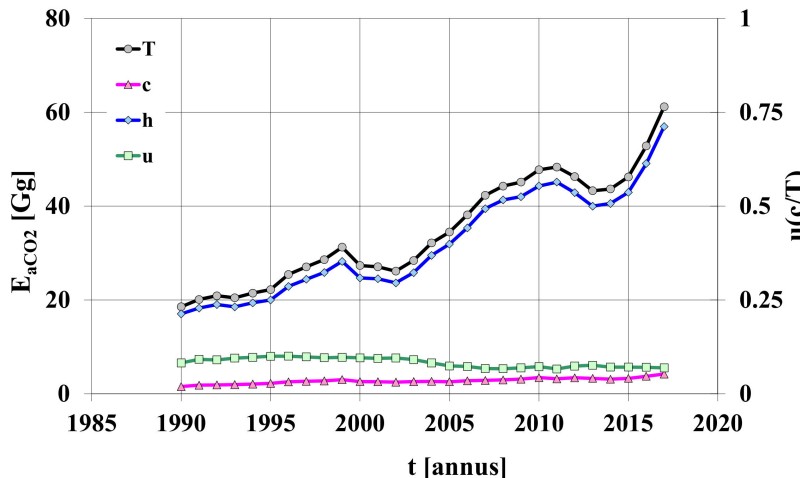

**Figure 5.** Annual emission of carbon dioxide—$E_{aCO2}$: national (total)—T, from car engines at cold-start—c, from car engines at warm-up—h and the share of additional emissions during engine heating in total emissions—u(c/T).

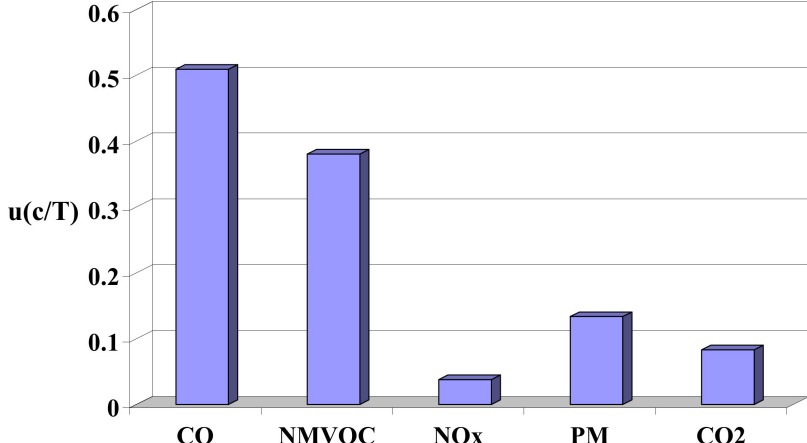

**Figure 6.** Share of additional annual emissions of pollutants during engine warm-up in the total annual emissions averaged over all inventory years.

Figures 1–5 show annual pollutant emissions, i.e., total emission at a national level, emissions from the engines analyzed during cold-start and warm-up, and Figure 6 shows the share of the additional emission in the heating time of engines in the total emission.

In the study period (1990–2017), national annual emissions of carbon monoxide and non-methane volatile organic compounds were decreasing ever after 1999, regardless of a boost in the use of motor vehicles. Emission reductions were a result of the technical advances made in the design of combustion engines and their exhaust after-treatment systems [15,17,24,28], as well as beneficial changes in the composition of the vehicle population in the country towards using vehicles with modern technological solutions, including with hybrid powertrains and, above all, electric vehicles [7,8].

The national annual emission of nitrogen oxides and particulate matter showed upward trends, with fluctuations in the years 1999–2005. The progress in the reduction in nitrogen oxides and particulate matter emissions was evidently lesser when compared to carbon monoxide and non-methane volatile organic compounds [15,17,24,28].

The observed increase in the national annual carbon dioxide emission was primarily associated with a fuel consumption increase as a result of intensive development of the automotive industry in Poland, including changes in the structure of vehicles in use-trends in the use of larger vehicles [7,8].

Figure 6 shows a set of shares of additional annual emission in the time of engine heating in the total annual emission, averaged for all inventory years, for all the analyzed pollutants.

Carbon monoxide and volatile organic compounds emissions are most sensitive to the thermal status of internal combustion engines. In the case of nitrogen oxides' emission from the engine during warm-up, low engine temperatures support low nitrogen oxides' emission [9,10,13,15,24]; however, then the effectiveness of the exhaust after-treatment system is decreased [10–13,16,19–21,23]. In the event of using the catalyst after-treatment, exhaust gas purification from particulate matter is slightly delayed—therefore, the effect of combustion engine warm-up on particulate matter emission is lesser when compared to that observed for nitrogen oxides.

The results of the simulation tests with the use of INFRAS AG software are presented in Figures 7–11.

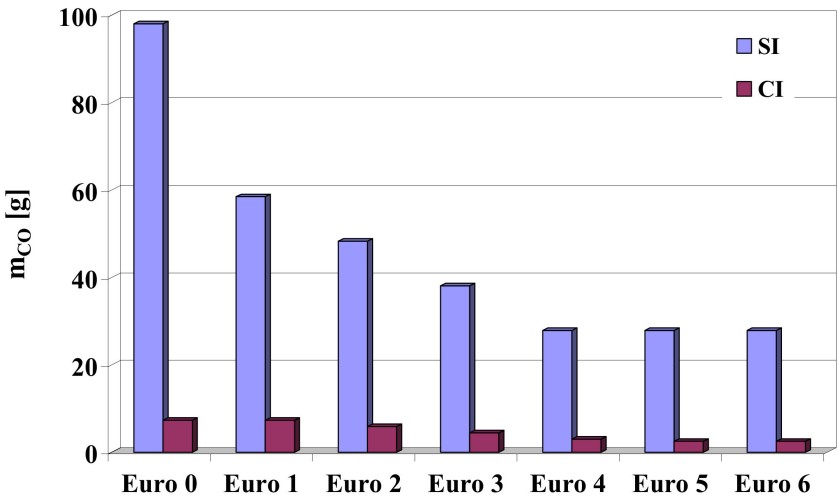

**Figure 7.** Additional carbon monoxide emission—$m_{CO}$, during warm-up of the spark-ignition engine—SI or the compression-ignition engine—CI of passenger cars of Euro 0–Euro 6 pollutant emission categories, under the model conditions for vehicle traffic.

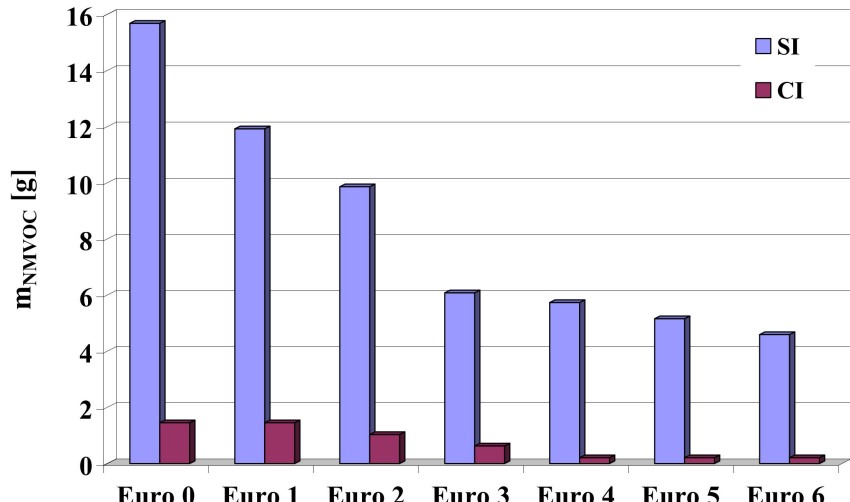

**Figure 8.** Additional emission of non-methane volatile organic compounds—$m_{NMVOC}$, during warm-up of the spark-ignition engine—SI or the compression-ignition engine—CI of passenger cars of Euro 0—Euro 6 pollutant emission categories, under the model conditions for vehicle traffic.

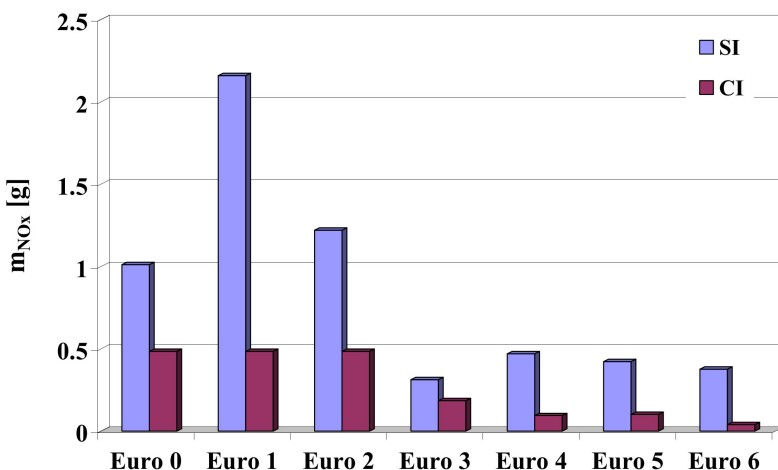

**Figure 9.** Additional emission of nitrogen oxides—$m_{NOx}$, during warm-up of the spark-ignition engine—SI or the compression-ignition engine—CI of passenger cars of Euro 0–Euro 6 pollutant emission categories, under the model conditions for vehicle traffic.

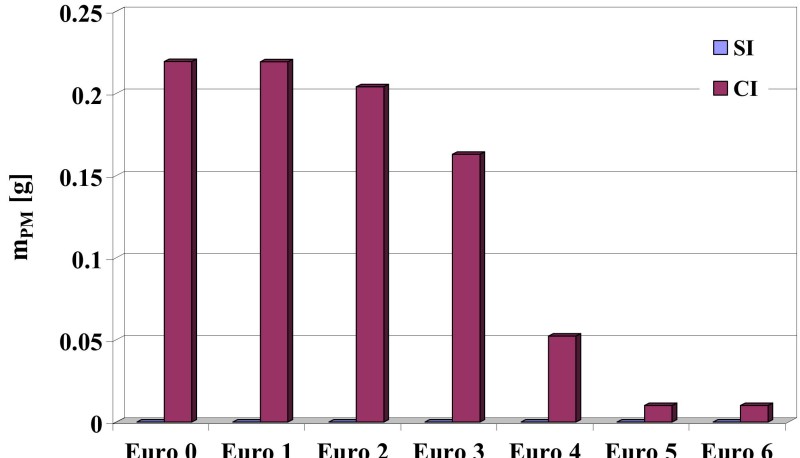

**Figure 10.** Additional emission of particulate matter—$m_{PM}$, during warm-up of the spark-ignition engine—SI or the compression-ignition engine—CI of passenger cars of Euro 0–Euro 6 pollutant emission categories, under the model conditions for vehicle traffic.

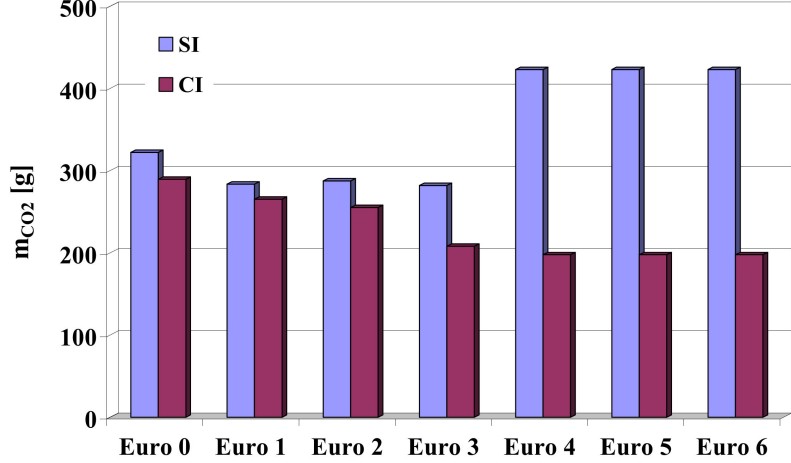

**Figure 11.** Additional emission of carbon dioxide—$m_{CO2}$, during warm-up of the spark-ignition engine—SI or the compression-ignition engine—CI of passenger cars of Euro 0–Euro 6 pollutant emission categories, under the model conditions for vehicle traffic.

These figures show additional pollutant emissions during engine warm-up for Euro 0–Euro 6 passenger cars with spark-ignition or compression-ignition engines, under the model vehicle movement conditions described in the methodology of analysis.

The test results obtained in the present study unequivocally confirm that additional pollutant emissions during engine warm-up are much higher in the case of spark-ignition engines as compared to compression-ignition engines [9,10,13,14,19–21,23]. However, this does not apply to particulate matter emission from spark-ignition engines, which is practically negligible, with the exception of direct-injection engines (in this case, the number of very fine particulates is very important, the so-called nanoparticles) [14,23].

Figures 12–16 show specific distance emissions from hot spark-ignition engines and compression-ignition engines of passenger cars of Euro 0–Euro 6 pollutant emission categories, under model vehicle traffic conditions.

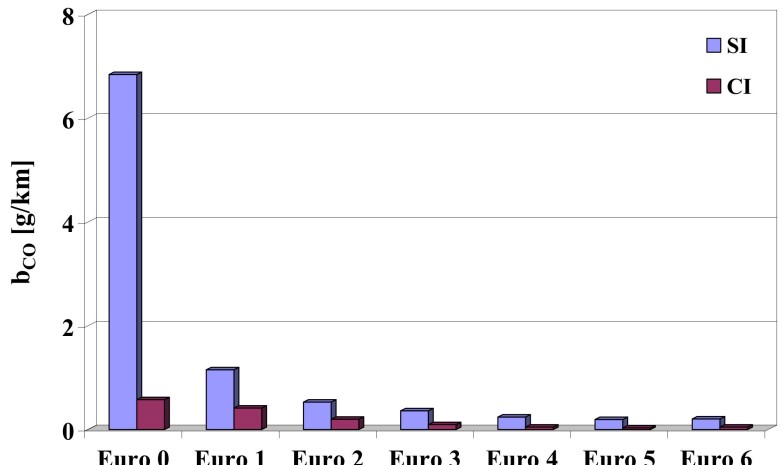

**Figure 12.** Specific distance emission of carbon monoxide—$b_{CO}$, from hot passenger car engines of Euro 0–Euro 6 pollutant emission categories, spark-ignition engines—SI and compression-ignition engines—CI, under the model vehicle traffic conditions.

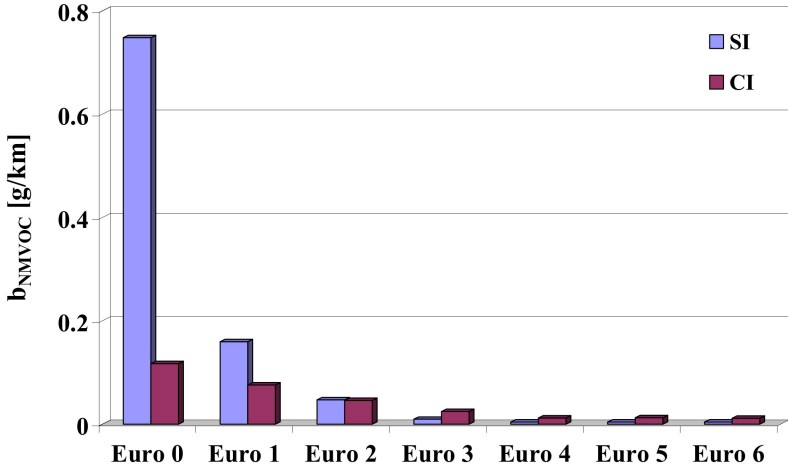

**Figure 13.** Specific distance emission of non-methane volatile organic compounds—$b_{NMVOC}$, from hot passenger car engines of Euro 0–Euro 6 pollutant emission categories, spark-ignition engines—SI and compression-ignition engines—CI, under the model vehicle traffic conditions.

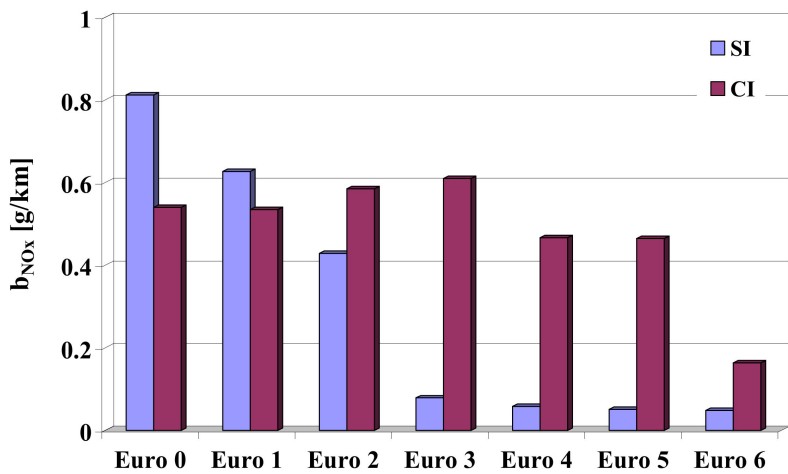

**Figure 14.** Specific distance emission of nitrogen oxides—$b_{NOx}$, from hot passenger car engines of Euro 0–Euro 6 pollutant emission categories, spark-ignition engines—SI and compression-ignition engines—CI, under the model vehicle traffic conditions.

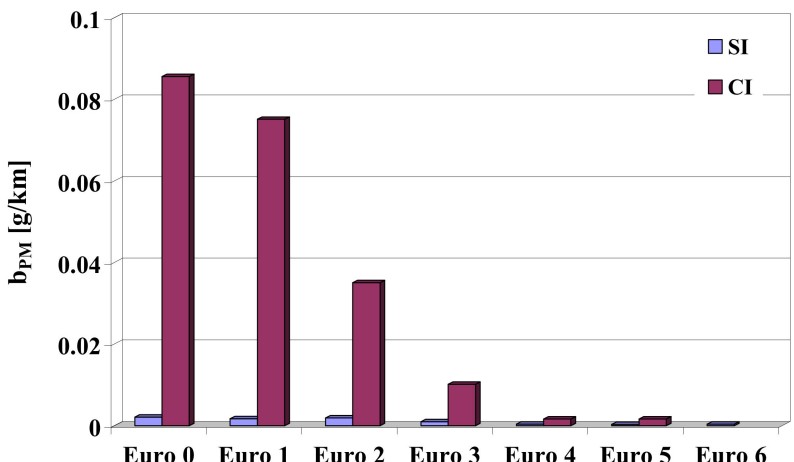

**Figure 15.** Specific distance emission of particulate matter—$b_{PM}$, from hot passenger car engines of Euro 0–Euro 6 pollutant emission categories, spark-ignition engines—SI and compression-ignition engines—CI, under the model vehicle traffic conditions.

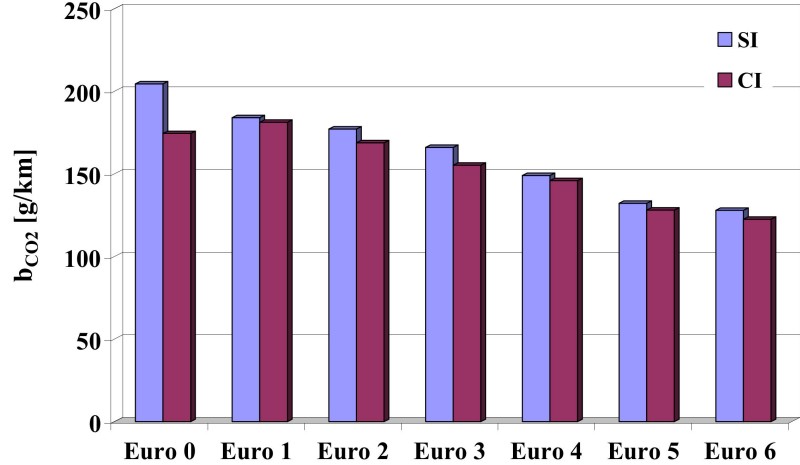

**Figure 16.** Specific distance emission of carbon dioxide—$b_{CO2}$, from hot passenger car engines of Euro 0–Euro 6 pollutant emission categories, spark-ignition engines—SI and compression-ignition engines—CI, under the model vehicle traffic conditions.

The results obtained evidently showed that the technical progress in the reduction in emissions from passenger car engines was by far the greatest for carbon monoxide and volatile organic compounds, especially in the case of spark-ignition engines [2,15,17,24]. There was also observed a great progress in the reduction in particulate matter emission from compression-ignition engines [2,15,17,24]. Progress in reducing nitrogen oxide emission is much more difficult. Similarly, the overall progress in the reduction in carbon dioxide emission into the atmosphere is not easy as well, due to limitations associated with the second law of thermodynamics, and a narrow possibility to increase engine thermal efficiency and thus—consequently—to reduce fuel consumption [10].

Figures 17–21 show the shares of additional pollutant emissions in total emissions, for passenger cars of six pollutant emission categories (Euro 0–Euro 6), during warm-up of spark-ignition engines and compression-ignition engines, under model vehicle movement traffic conditions.

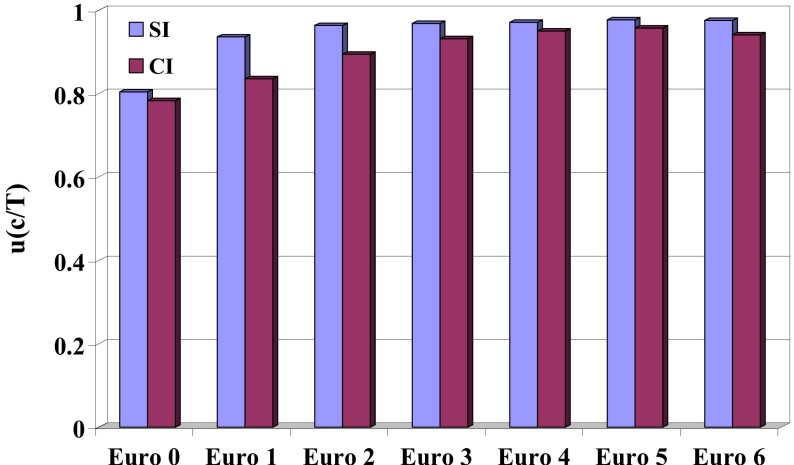

**Figure 17.** Share of additional carbon oxide emission in total emission—u(c/T), from passenger cars of emission categories Euro 0–Euro 6 during warm-up of spark-ignition engines—SI and pollutant compression-ignition engines—CI, under model vehicle traffic conditions.

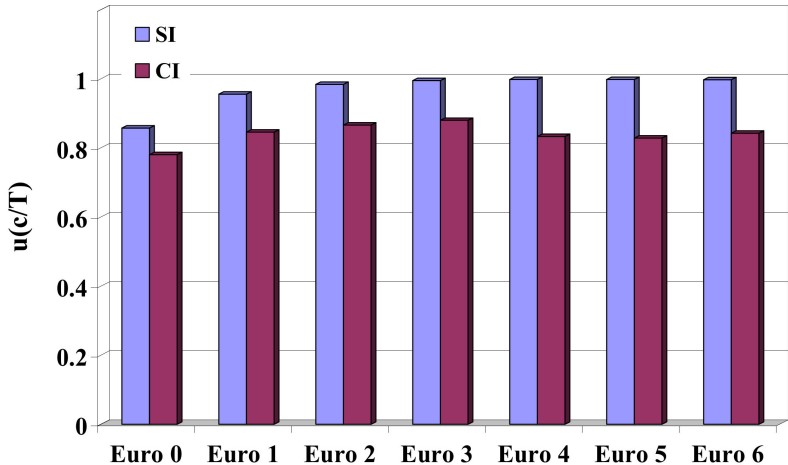

**Figure 18.** Share of additional non-methane volatile organic compounds emission in total emission—u(c/T), from passenger cars of pollutant emission categories Euro 0–Euro 6, during warm-up of spark-ignition engines—SI and compression-ignition engines—CI, under model vehicle traffic conditions.

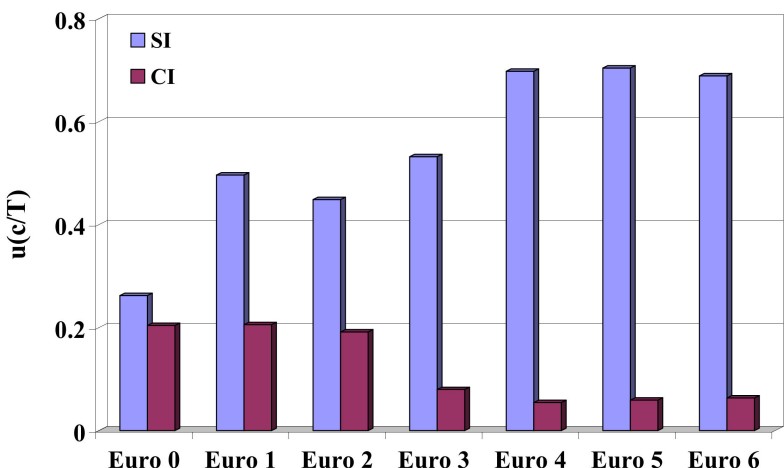

**Figure 19.** Share of additional nitrogen oxides emission in total emission—u(c/T), from passenger cars of pollutant emission categories Euro 0–Euro 6, during warm-up of spark-ignition engines—SI and compression-ignition engines—CI, under model vehicle traffic conditions.

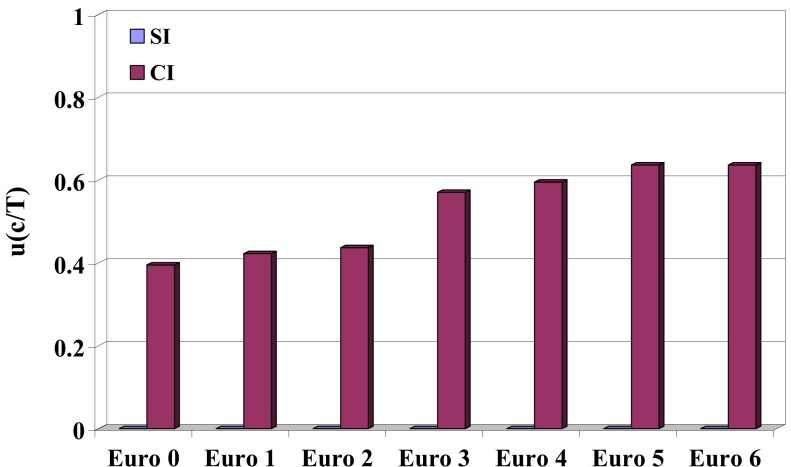

**Figure 20.** Share of additional particulate matter emission in total emission—u(c/T), from passenger cars of pollutant emission categories Euro 0–Euro 6, during warm-up of spark-ignition engines—SI and compression-ignition engines—CI, under model vehicle traffic conditions.

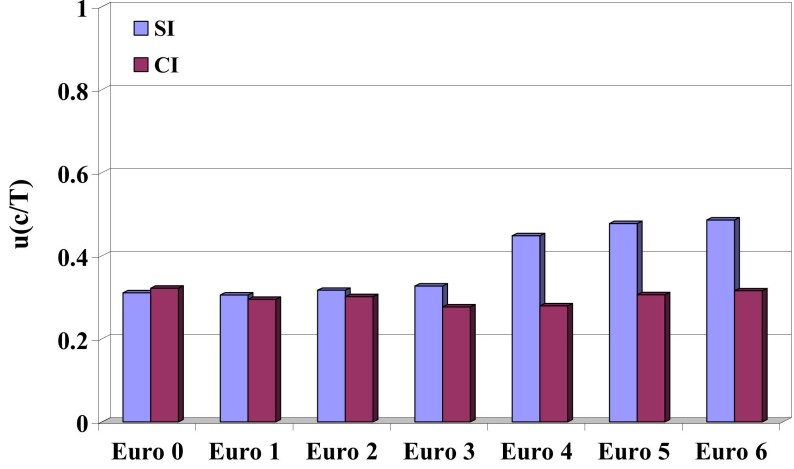

**Figure 21.** Share of additional carbon dioxide emission in total emission—u(c/T), from passenger cars of pollutant emission categories Euro 0–Euro 6, during warm-up of spark-ignition engines—SI and compression-ignition engines—CI, under model vehicle traffic conditions.

As expected, carbon monoxide and organic compounds showed definitely the largest share of additional emissions in total annual emissions during spark-ignition engine warm-up. The smallest additional emission share was observed in the case of nitrogen oxides from compression-ignition engines [3,4,7]. In the case of carbon dioxide additional emission, there were observed relatively small shares: (30 ÷ 45)% for spark-ignition engines and 30% for compression-ignition engines.

## 4. Summary—Discussion of the Results and Conclusions

The following conclusions can be drawn from the results of the present study:

1.  Even with the ongoing considerable intensification of road transport in Poland, national annual emissions of hazardous pollutants, both total and from combustion engines under warm-up conditions, show evident decreasing trends in the case of carbon monoxide and volatile organic compounds. The annual emissions of nitrogen oxides and particulate matter have remained stable (Figures 1–5) [7,8]. Emission reductions are associated with the technological advancement of the motor industry, and particularly with the use of modern solutions to reduce pollutant emissions from vehicles. The upward trend observed in annual carbon dioxide emission results from the intensification of road transport and, as a consequence, increased fuel consumption. The transport sector has strived hard to reduce fuel use and carbon dioxide emission; however the effectiveness of the efforts undertaken is limited due to the laws of nature that affect the thermal efficiency of internal combustion engine circuits and, first of all, by the second law of thermodynamics.

2.  The share of annual pollutant emissions from car engines under cold-start conditions in total annual emissions remains almost constant. Emissions of carbon monoxide and volatile organic compounds constitute the largest shares—about 50%, whereas the particulate matter share is less than 15%, carbon dioxide less than 10% and nitrogen oxides less than 5%.

3.  The additional emission of pollutants from passenger cars' combustion engines during engine warm-up and vehicle movement under urban conditions (Figures 7–11) shows, with some exceptions, a decreasing trend, which results from the technological advancements of the motor industry. This does not apply to the emissions of carbon dioxide (Conclusion 1) and of nitrogen oxides from passenger cars designated to the Euro 1 pollutant emission category and equipped with spark-ignition engines which use multifunctional catalytic converters, which are efficient in reducing pollutants in exhaust gases but sensitive to operating temperatures in terms of reducing nitrogen oxides. The reduction in additional emissions as a result of the warm-up of internal combustion engines shows the strongest trend in the case of compounds with reducing properties in oxygen-containing environments: carbon monoxide and non-methane volatile organic compounds.

4.  The reduction in the specific distance emission of pollutants from combustion engines of passenger cars designated to the explicit pollutant emission categories is very clear—the strongest for carbon monoxide and non-methane volatile organic compounds (Figures 12–16).

5.  In total emissions, there was observed a very high share of additional emissions from combustion engines during warm-up under the conditions of passenger car urban driving (Figures 17–21), especially for spark-ignition engines: for carbon monoxide and non-methane volatile organic compounds, it was about 90%, and lesser for nitrogen oxides, especially in the case of compression-ignition engines, as well as for particulate matter in the case of compression-ignition engines. The carbon dioxide share constituted about 30%.

In general, the results obtained allow for the conclusion that, on a global scale, there is possible substantial progress in the reduction in pollutant emissions from automotive combustion engines, especially during engine warm-up. Therefore, further works towards the improvement of internal combustion engines are necessary, e.g., on the accel-

eration of the heating of catalytic exhaust after-treatment systems and on the long-term maintenance of the thermal status of the engine after stopping with the use of thermal accumulators [29–32].

**Supplementary Materials:** The following are available online at: https://www.mdpi.com/article/10.3390/app11199084/s1, Categories of vehicles that are defined in the COPERT 5 software, Table S1: Annual carbon monoxide emission—EaCO: national (total)—T, from car engines at cold-start—c, from car engines at warm-up—h and the share of additional emissions during engine heating in total emissions—u(c/T), Table S2: Annual emission of non-methane volatile organic compounds: EaNMVOC: national (total)—T, from car engines at cold-start—c, from car engines at warm-up—h and the share of additional emissions during engine heating in total emissions—u(c/T), Table S3: Annual emission of nitrogen oxides—EaNOx: national (total)—T, from car engines at cold-start—c, from car engines at warm-up—h and the share of additional emissions during engine heating in total emissions—u(c/T), Table S4: Annual emission of particulate matter—EaPM: national (total)—T, from car engines at cold-start—c, from car engines at warm-up—h and the share of additional emissions in the time of engine heating in total emissions—u(c/T), Table S5: Annual emission of carbon dioxide—$E_{aCO2}$: national (total)—T, from car engines at cold-start—c, from car engines at warm-up—h and the share of additional emissions during engine heating in total emissions—u(c/T), Table S6: Share of additional annual emissions of pollutants during engine warm-up in the total annual emissions averaged over all inventory years, Table S7: Additional carbon monoxide emission—mCO, during warm-up of the spark-ignition engine—SI or the compression-ignition engine—CI of passenger cars of Euro 0–Euro 6 pollutant emission categories, under the model conditions for vehicle traffic, Table S8: Additional emission of non-methane volatile organic compounds—mNMVOC, during warm-up of the spark-ignition engine—SI or the compression-ignition engine—CI of passenger cars of Euro 0–Euro 6 pollutant emission categories, under the model conditions for vehicle traffic, Table S9: Additional emission of nitrogen oxides—mNOx, during warm-up of the spark-ignition engine—SI or the compression-ignition engine—CI of passenger cars of Euro 0–Euro 6 pollutant emission categories, under the model conditions for vehicle traffic, Table S10: Additional emission of particulate matter—mPM, during warm-up of the spark-ignition engine—SI or the compression-ignition engine—CI of passenger cars of Euro 0–Euro 6 pollutant emission categories, under the model conditions for vehicle traffic, Table S11: Additional emission of carbon dioxide—$m_{CO2}$, during warm-up of the spark-ignition engine—SI or the compression-ignition engine—CI of passenger cars of Euro 0–Euro 6 pollutant emission categories, under the model conditions for vehicle traffic, Table S12: Specific distance emission of carbon monoxide—bCO from hot passenger car engines of Euro 0–Euro 6 pollutant emission categories, spark-ignition engines—SI and compression-ignition engines—CI, under the model vehicle traffic conditions, Table S13: Specific distance emission of non-methane volatile organic compounds—bNMVOC from hot passenger car engines of Euro 0–Euro 6 pollutant emission categories, spark-ignition engines—SI and compression-ignition engines—CI, under the model vehicle traffic conditions, Table S14: Specific distance emission of nitrogen oxides—bNOx from hot passenger car engines of Euro 0–Euro 6 pollutant emission categories, spark-ignition engines—SI and compression-ignition engines—CI, under the model vehicle traffic conditions, Table S15: Specific distance emission of particulate matter—bPM from hot passenger car engines of Euro 0–Euro 6 pollutant emission categories, spark-ignition engines—SI and compression-ignition engines—CI, under the model vehicle traffic conditions, Table S16: Specific distance emission of carbon dioxide—$b_{CO2}$ from hot passenger car engine of Euro 0–Euro 6 pollutant emission categories, spark-ignition engines—SI and compression-ignition engines—CI, under the model vehicle traffic conditions, Table S17: Share of additional carbon oxide emission in total emission—u(c/T), from passenger cars of emission categories Euro 0–Euro 6 during warm-up of spark-ignition engines—SI and pollutant compression-ignition engines—CI, under model vehicle traffic conditions, Table S18: Share of additional non-methane volatile organic compounds emission in total emission—u(c/T), from passenger cars of pollutant emission categories Euro 0–Euro 6, during warm-up of spark-ignition engines—SI and compression-ignition engines—CI, under model vehicle traffic conditions, Table S19: Share of additional nitrogen oxides emission in total emission—u(c/T), from passenger cars of pollutant emission categories Euro 0–Euro 6, during warm-up of spark-ignition engines—SI and compression-ignition engines—CI, under model vehicle traffic conditions, Table S20: Share of additional particulate matter emission in total emission—u(c/T), from passenger cars of pollutant emission categories Euro 0–Euro 6, during warm-up of spark-ignition engines—SI and compression-

ignition engines—CI, under model vehicle traffic conditions, Table S21: Share of additional carbon dioxide emission in total emission—u(c/T), from passenger cars of pollutant emission categories Euro 0–Euro 6, during warm-up of spark-ignition engines—SI and compression-ignition engines—CI, under model vehicle. traffic conditions.

**Author Contributions:** Conceptualization, K.B. and Z.C.; methodology, K.B., Z.C., K.S. and M.Z.-L.; Software, K.B. and Z.C.; Validation, K.B., Z.C., K.S. and M.Z.-L.; Formal analysis, K.B., Z.C. and H.S.; Investigation, K.B., Z.C., K.S. and M.Z.-L.; Resources, K.B., Z.C., K.S. and M.Z.-L.; Data curation, K.B., Z.C., H.S. and M.Z.-L.; Writing—original draft preparation, Z.C. and H.S.; Writing—review and editing, K.B., Z.C. and H.S.; Visualization, Z.C. and H.S.; Supervision, Z.C. and K.S.; Project administration, K.S.; Funding acquisition, K.S. All authors have read and agreed to the published version of the manuscript.

**Funding:** The APC was funded by the Institute of Environmental Protection—National Research Institute.

**Institutional Review Board Statement:** Not applicable.

**Informed Consent Statement:** Not applicable.

**Data Availability Statement:** Not applicable.

**Conflicts of Interest:** The authors declare no conflict of interest.

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
