# Peer review of "Influence of the Thermal State of Vehicle Combustion Engines on the Results of the National Inventory of Pollutant Emissions"

_applsci, doi:10.3390/app11199084_

Round 1
Reviewer 1 Report
This is an interesting paper to be read, overall, it is well written and provided valuable information for evaluation of the environment impact caused by automotive industry, the suggested modifications are listed below,
- The reason of national annual NOx and PM emission increment should be included.
- It is suggested to replace 'positive ignition engines' by 'spark ignition engines'.
- The reason of additional emission of CO2 increment during SI engine warm-up should be included.
- Fig.16 presents a downward trend of CO2 emission which is contradictory with Fig.5, therefore the reason of this phenomenon should be presented.
Author Response
Manuscript (id applsci-1365206) entitled:
Influence of the thermal state of vehicle combustion engines on the results of the national inventory of pollutant emissions
Dear Reviewer No. 1!
Thank You very much for Your thorough acknowledgement with the manuscript. Below we presented the answers for all Your remarks.
With kind regards,
Authors.
Detailed remark No. 1 of Reviewer No. 1:
- The reason of national annual NOx and PM emission increment should be included.
Response to detailed remark No. 1 of Reviewer No. 1:
This is mainly due to the dynamic growth in the number of motor vehicles. At the same time, technical progress in reducing nitrogen oxides and particulate emissions is much slower than in the cases of carbon monoxide and organic compounds.
Detailed remark No. 2 of Reviewer No. 1:
- It is suggested to replace 'positive ignition engines' by 'spark ignition engines'.
Response to detailed remark No. 2 of Reviewer No. 1:
Thank You for this remark. Your suggestion has been included in the manuscript.
Detailed remark No. 3 of Reviewer No. 1:
- The reason of additional emission of CO2 increment during SI engine warm-up should be included.
Response to detailed remark No. 3 of Reviewer No. 1:
It seems that this would be trivial information, because it is widely known that during the cold start of the engine, fuel consumption is increased, which means increased carbon dioxide emissions. About fuel consumption is written in the article.
Detailed remark No. 4 of Reviewer No. 1:
- Fig.16 presents a downward trend of CO2 emission which is contradictory with Fig.5, therefore the reason of this phenomenon should be presented.
Response to detailed remark No. 4 of Reviewer No. 1:
Figure 5 shows the annual emission, and Figure 16 shows specific distance emission. Specific distance emission is declining due to technological progress, and annual emission is increasing as the number of cars increases.

Reviewer 2 Report
The authors provided a typescript on the subject “Influence of the thermal state of vehicle combustion engines on the results of the national inventory of pollutant emissions”. The authors presented the introduction in detail. A literature review has been properly prepared in the introduction.
The methodology could be described in more detail. The COPERT 5 software has several different variants. Please fill in which version was used for testing. The authors write that they conducted the simulation according to INFRAS AG software. Who is the manufacturer of the software and whether they are dedicated to the research? Please show the differences why two different software were used. Can the obtained results be compared with each other?
The presentation of the research results in the form of drawings is, to put it mildly, difficult to interpret. It is recommended to present the obtained test results in tables. The lines show the pollution indicators and the columns show the results from the software.
Currently, it is difficult to find valid conclusions from the experiments carried out.
Author Response
Manuscript (id applsci-1365206) entitled:
Influence of the thermal state of vehicle combustion engines on the results of the national inventory of pollutant emissions
Dear Reviewer No. 2!
Thank You very much for Your thorough acknowledgement with the manuscript. Below we presented the answers for all Your remarks.
With kind regards,
Authors.
Detailed remark No. 1 of Reviewer No. 2:
- The methodology could be described in more detail. The COPERT 5 software has several different variants. Please fill in which version was used for testing. The authors write that they conducted the simulation according to INFRAS AG software. Who is the manufacturer of the software and whether they are dedicated to the research? Please show the differences why two different software were used. Can the obtained results be compared with each other?
Response to detailed remark No. 1 of Reviewer No. 2:
The article is extensive. The research methodology is described in detail in the literature to which we refer. A detailed description of the research methodology is the subject of a monograph, for example: Chłopek, Z.: Modelowanie procesów emisji spalin w warunkach eksploatacji trakcyjnej silników spalinowych. (Modeling of exhaust emission processes under traction operation of internal combustion engines). Prace Naukowe. Seria "Mechanika" z. 173. Oficyna Wydawnicza Politechniki Warszawskiej. Warszawa 1999. (Habilitation dissertation - In Polish).
Detailed remark No. 2 of Reviewer No. 2:
- The presentation of the research results in the form of drawings is, to put it mildly, difficult to interpret. It is recommended to present the obtained test results in tables. The lines show the pollution indicators and the columns show the results from the software.
Response to detailed remark No. 2 of Reviewer No. 2:
It is difficult to agree with this remark as the role of graphs is to better describe the results in a graphical way. It is not possible to do it in table. The information about results could be depicted in tables. They are put in MS Excel software, to be able to build graphs. However, it would unnecessarily extent the volume of the manuscript, not increasing the scientific value of the manuscript.
Detailed remark No. 3 of Reviewer No. 2:
- Currently, it is difficult to find valid conclusions from the experiments carried out.
Response to detailed remark No. 3 of Reviewer No. 2:
The paper proves the share of pollutant emissions during the heating up of engines after start-up, when the efficiency of catalytic reactors is very low. Therefore, it was concluded in the summary that it is necessary to intensify the work, e.g. over the acceleration of heating of catalytic reactors. It is a practical conclusion, and the paper presents the scale of the problem that justifies this conclusion.

Round 2
Reviewer 2 Report
I am asking the authors to provide the full results of the research (MS Excel) in supplementary materials.
Author Response
Manuscript (id: applsci-1365206) entitled:
Influence of the thermal state of vehicle combustion engines on the results of the national inventory of pollutant emissions
Dear Reviewer No. 2!
Thank You very much for thorough acknowledgement with the manuscript. We have included Your remark as mentioned below. Additionally, we have replaced Figures 1 – 5. New parts of the text have been also highlighted by red colour. All the changes are of course visible thanks to turning on “changes tracking” functionality in MS Word.
With kind regards,
Authors.
Comments and Suggestions for Authors
General remark of Reviewer No. 2:
I am asking the authors to provide the full results of the research (MS Excel) in supplementary materials.
Response to Reviewer's general remark No. 2:
Thank You very much for this remark. Supplemental materials including full results of the research in MS Excel have been added (as separate MS Excel file), together with division of vehicles into categories applied in COPERT 5 software.
